# *Spiroplasma* Infection among Ixodid Ticks Exhibits Species Dependence and Suggests a Vertical Pattern of Transmission

**DOI:** 10.3390/microorganisms9020333

**Published:** 2021-02-08

**Authors:** Shohei Ogata, Wessam Mohamed Ahmed Mohamed, Kodai Kusakisako, May June Thu, Yongjin Qiu, Mohamed Abdallah Mohamed Moustafa, Keita Matsuno, Ken Katakura, Nariaki Nonaka, Ryo Nakao

**Affiliations:** 1Laboratory of Parasitology, Department of Disease Control, Faculty of Veterinary Medicine, Graduate School of Infectious Diseases, Hokkaido University, N 18 W 9, Kita-ku, Sapporo 060-0818, Japan; s.ogata@vetmed.hokudai.ac.jp (S.O.); wessam@czc.hokudai.ac.jp (W.M.A.M.); kusaki@vmas.kitasato-u.ac.jp (K.K.); winterzun@gmail.com (M.J.T.); m.abdallah@vetmed.hokudai.ac.jp (M.A.M.M.); kenkata@vetmed.hokudai.ac.jp (K.K.); nnonaka@vetmed.hokudai.ac.jp (N.N.); 2Laboratory of Veterinary Parasitology, School of Veterinary Medicine, Kitasato University, Towada, Aomori 034-8628, Japan; 3Hokudai Center for Zoonosis Control in Zambia, School of Veterinary Medicine, The University of Zambia, P.O. Box 32379, Lusaka 10101, Zambia; yongjin_qiu@czc.hokudai.ac.jp; 4Department of Animal Medicine, Faculty of Veterinary Medicine, South Valley University, Qena 83523, Egypt; 5Unit of Risk Analysis and Management, Research Center for Zoonosis Control, Hokkaido University, N 20 W 10, Kita-ku, Sapporo 001-0020, Japan; matsuk@czc.hokudai.ac.jp; 6International Collaboration Unit, Research Center for Zoonosis Control, Hokkaido University, N 20 W 10, Kita-ku, Sapporo 001-0020, Japan

**Keywords:** *Haemaphysalis*, *Ixodes*, *Spiroplasma*, symbionts, ticks, Japan

## Abstract

Members of the genus *Spiroplasma* are Gram-positive bacteria without cell walls. Some *Spiroplasma* species can cause disease in arthropods such as bees, whereas others provide their host with resistance to pathogens. Ticks also harbour *Spiroplasma*, but their role has not been elucidated yet. Here, the infection status and genetic diversity of *Spiroplasma* in ticks were investigated using samples collected from different geographic regions in Japan. A total of 712 ticks were tested for *Spiroplasma* infection by PCR targeting 16S rDNA, and *Spiroplasma* species were genetically characterized based on 16S rDNA, ITS, *dnaA*, and *rpoB* gene sequences. A total of 109 samples originating from eight tick species were positive for *Spiroplasma* infection, with infection rates ranging from 0% to 84% depending on the species. A linear mixed model indicated that tick species was the primary factor associated with *Spiroplasma* infection. Moreover, certain *Spiroplasma* alleles that are highly adapted to specific tick species may explain the high infection rates in *Ixodes ovatus* and *Haemaphysalis kitaokai*. A comparison of the alleles obtained suggests that horizontal transmission between tick species may not be a frequent event. These findings provide clues to understand the transmission cycle of *Spiroplasma* species in wild tick populations and their roles in host ticks.

## 1. Introduction

Members of the genus *Spiroplasma* are Gram-positive bacteria without cell walls. They are known as symbionts of arthropods and plants and are classified into three major clades based on the 16S ribosomal RNA gene (rDNA) sequence: Ixodetis, Citri-Chrysopicola-Mirum (CCM), and Apis [1,2]. *Spiroplasma* is one of the most common endosymbionts with a wide range of hosts, including insects, arachnids, crustaceans, and plants [3]. It is estimated that 5–10% of insect species harbor this symbiont group [2,4].

*Spiroplasma* has a wide range of fitness effects and transmission strategies [2,5,6,7,8,9,10,11,12,13,14,15,16,17]. Some *Spiroplasma* species affect the sex ratio by inducing male killing in hosts such as flies, butterflies, and ladybird beetles [7,8,9,10]. Several *Spiroplasma* species are known to cause disease in arthropods such as bees and plants [6,17,18]. On the other hand, some flies infected with *Spiroplasma* can develop resistance to other pathogens [5,10,11,12]. A wide range of symbiotic relationships involving *Spiroplasma* have been observed [5,7,8,14,15,16]. The rapid spread of *Spiroplasma* infection in fruit fly natural populations has been reported in some areas of North America, and this phenomenon has been confirmed in laboratory settings [19]. This characteristic of *Spiroplasma* is not only biologically interesting, but also useful for symbiotic control applications among host individuals [20].

Ticks have long been studied, since they transmit a variety of pathogens to humans and animals. *Spiroplasma mirum* is the first reported tick-associated *Spiroplasma,* which was obtained from *Haemaphysalis leporispalustris* in the United States in 1982 during the search for rickettsiae in ticks [21]. Another species, *S. ixodetis*, was isolated from *Ixodes pacificus* in the United States in 1981 [22]. Thus far, these two species are the only validated *Spiroplasma* species detected in ticks. Nevertheless, several alleles or putative new species of *Spiroplasma* have been found in various tick species such as *I. arboricola, I. frontalis, I. ovatus, I. persulcatus, I. ricinus, I. uriae, Dermacentor marginatus, Rhipicephalus annulatus, R. decoloratus, R. geigyi,* and *R. pusillus* [23,24,25,26,27,28,29,30].

In Japan, 46 tick species belonging to seven genera (*Amblyomma*, *Argas*, *Dermacentor*, *Rhipicephalus, Haemaphysalis*, *Ixodes*, and *Ornithodoros*) have been recorded [11,12]. Several tick-borne diseases such as Lyme disease, relapsing fever, Japanese spotted fever, severe fever with thrombocytopenia syndrome, and tick-borne encephalitis are endemic [31]. Taroura et al. first detected *Spiroplasma* DNA in questing *I. ovatus* ticks captured in several prefectures [24]. Subsequently, a microbiome study revealed the presence of *Spiroplasma* in the salivary glands of *I. ovatus* and *I. persulcatus* [23]. More recently, several *Spiroplasma* isolates were obtained by incubating the homogenates of *I. monospinosus*, *I. persulcatus*, and *H. kitaokai* with tick and mosquito cells [32]. These studies collectively indicate that there is a close relationship between *Spiroplasma* and ticks in Japan; however, no comprehensive studies have been conducted to determine the genetic diversity and prevalence of tick-associated *Spiroplasma*.

The aim of this study was to identify and genetically characterize *Spiroplasma* in different tick species in Japan. A linear mixed model (LMM) was developed to resolve the correlation among several extrinsic and intrinsic factors associated with *Spiroplasma* infection in ticks.

## 2. Materials and Methods

### 2.1. Sample Collection

Ticks were collected by flagging the vegetation during the period of tick activity (between April 2013 and August 2018) at 112 different sampling sites in 19 different prefectures in Japan. The sampling sites were classified into nine geographical blocks: Hokkaido (Hokkaido prefecture), Tohoku (Yamagata and Fukushima prefectures), Kanto (Chiba prefecture), Chubu (Nagano and Shizuoka prefectures), Kinki (Mie, Nara, and Wakayama prefectures), Chugoku (Hiroshima and Shimane prefectures), Shikoku (Kagawa, Ehime, and Kochi prefectures), Kyushu (Nagasaki, Kumamoto, Miyazaki, and Kagoshima prefectures), and Okinawa (Okinawa prefecture). All collected ticks were transferred to Petri dishes and preserved in an incubator at 16 °C until use.

### 2.2. Identification of Tick Species

Tick species were identified morphologically under a stereomicroscope according to standard morphological keys [33,34]. A total of 712 adult ticks from four genera were examined in this study. These included two species in the genus *Amblyomma* (*A. geoemydae*, *n* = 3; *A. testudinarium*, *n* = 26), one species in the genus *Dermacentor* (*D. taiwanensis*, *n* = 9), 10 species in the genus *Haemaphysalis* (*H. concinna*, *n* = 2; *H. cornigera*, *n* = 1; *H. flava*, *n* = 65; *H. formosensis*, *n* = 83; *H. hystricis*, *n* = 60; *H. japonica*, *n* = 20; *H. kitaokai*, *n* = 78; *H. longicornis*, *n* = 106; *H. megaspinosa*, *n* = 66; *H. yeni*, *n* = 1), and seven species in the genus *Ixodes* (*I. monospinosus*, *n* = 21; *I. nipponensis*, *n* = 3; *I. ovatus*, *n* = 80; *I. pavlovsky*, *n* = 26; *I. persulcatus*, *n* = 55; *I. tanuki*, *n* = 1; *I. turdus*, *n* = 6).

### 2.3. DNA Extraction

The procedures for DNA extraction from individual ticks have been reported previously [35]. In brief, the surface of tick bodies was individually washed with 70% ethanol and sterilized phosphate-buffered solution (PBS). The whole tick bodies were homogenized in 100 μL of high-glucose Dulbecco’s modified Eagle’s medium (Gibco, Life Technologies, Grand Island, NY, USA) using Micro Smash MS100R (TOMY, Tokyo, Japan) for 30 s at 3000 rpm. DNA was extracted from 50 μL of the tick homogenate using the blackPREP Tick DNA/RNA Kit (Analytik Jena, Jena, Germany) according to the manufacturer’s protocol.

### 2.4. Detection of Spiroplasma in Ticks

To detect *Spiroplasma* DNA, PCR amplification of a sequence of approximately 1028 bp in the 16S rDNA was performed. The PCR was carried out in a 20 μL reaction mixture containing 10 μL of 2× Gflex PCR Buffer (Mg^2+^, dNTP plus), 400 nM of Tks Gflex™ DNA Polymerase (Takara Bio, Shiga, Japan), 400 nM of each primer, 1 μL of DNA template, and sterilized water. The reaction was performed at 94 °C for 1 min, followed by 45 cycles at 98 °C for 10 s, 60 °C for 30 s, and 68 °C for 45 s and a final step at 68 °C for 5 min. PCR products were electrophoresed on a 1.0% agarose gel. The DNA of a *Spiroplasma* species isolated from *I. persulcatus* in our previous study [23] and sterilized water were included in each PCR run as positive and negative controls, respectively. Primer sets used for each assay are shown in Table 1 [13,36]. The amplified PCR products were purified using ExoSAP-IT Express PCR Cleanup Reagent (Thermo Fisher Scientific, Tokyo, Japan). Sanger sequencing was performed using the BigDye Terminator version 3.1 Cycle Sequencing Kit (Applied Biosystems, Foster City, CA, USA) and the ABI Prism 3130xl Genetic Analyzer according to the manufacturer’ s instructions. Sequence data were assembled using ATGC software version 6.0.4 (GENETYX, Tokyo, Japan).

### 2.5. Molecular Characterization of Spiroplasma

To further characterize *Spiroplasma* in ticks, additional PCRs based on the 16S–23S rRNA intergenic transcribed spacer (ITS) region (301 bp), chromosomal replication initiator protein dnaA (*dnaA*) (515 bp), and RNA polymerase B (*rpoB*) genes (1703 bp) were performed with primers widely used for the characterization of *Spiroplasma* in arthropods [2,36]. These PCRs were performed for selected samples using the following criteria: (1) more than three samples (when available) were selected for each 16S rDNA allele; (2) the samples were selected from each tick species when the 16S rDNA allele was obtained from multiple tick species. The PCRs were carried out as described above, except that 56 and 52 °C were used as the annealing temperatures for ITS and *dnaA* PCRs, respectively. The primer sets used for each assay are shown in Table 1. All PCR amplicons were subjected to Sanger sequencing analysis. The sequences obtained were submitted to the DNA Data Bank of Japan (DDBJ) (http://www.ddbj.nig.ac.jp) under specific accession numbers (16S rDNA: LC592079–LC592113; ITS: LC592139–C592161; *dnaA*: LC592127–LC592138; *rpoB*: LC592114–LC592126).

### 2.6. Phylogenetic Analysis

Phylogenetic trees were constructed based on the partial sequences of 16S rDNA, *dnaA*, *rpoB* genes, and ITS region. The nucleotide sequences obtained were aligned with representative sequences of known *Spiroplasma* species available in GenBank as implemented in MEGA7 [30,37]. The reference sequences of ITS region of *S. ixodetis* were obtained by *de novo* assembly of Illumina raw reads of *Spiroplasma*-infected African monarch butterfly *Danaus chrysippus* deposited in the sequence read archives (SRA) of the NCBI with accession numbers of SRX3872086 and SRX3872088-SRX3872090 [38] using CLC Genomics Workbench v 20.0.4 (Qiagen, Hilden, Germany). Phylogenetic trees were constructed using maximum likelihood (ML) method with bootstrap tests of 1000 replicates. The sequence data of the evolutionary models were determined using the Akaike information criterion with MEGA7 [37].

### 2.7. Phylogenetic Analysis

*Spiroplasma* infection in ticks can be affected by various extrinsic and intrinsic factors. Here, the extrinsic factors included sampling district, city/town, season, month, and year variations, and the intrinsic factors were tick species and sex. First, multicollinearity among the explanatory variables was examined using pairwise correlations and the “VIF” function in R package [39] to determine whether multicollinearity was likely to influence LMM results. A correlation between several variables affecting *Spiroplasma* infection in tsetse flies was reported in a previous study [40]. To identify this possible correlation in ticks, we performed multiple correspondence analysis (MCA) using the “MCA” and “fviz_mca_var” functions in the R packages FactoMineR and Factoextra, respectively [41]. We used an LMM to resolve the correlation among the predictor variables associated with *Spiroplasma* infection in ticks. We fit the LMM with the predictor variables (sampling season, year, tick sex, and species) as the fixed effects with and without geographic location (district) as the random effect. This was followed by testing in additional LMMs using combinations of the predictor variables with district as the random effect variable and *Spiroplasma* infection as the response variable. We compared the effectiveness of the tested models with the Chi-square test using the “ANOVA” function in R software. Finally, the “lmer” function in the R package lme4 [42] was used for the selected LMM, with each detected *Spiroplasma* allele as the response variable.

## 3. Results

### 3.1. Infection Rate of Spiroplasma in Different Tick Species

In this study, 109 of 712 samples (15%) were positive for *Spiroplasma* infection. Among the 20 different tick species, eight tick species were positive for *Spiroplasma* infection, and the highest infection rate was observed in *I. ovatus* (84%; 67/80), followed by *H. kitaokai* (35%; 27/78), *I. turdus* (17%; 1/6), *I. persulcatus* (16%; 9/55), *D. taiwanensis* (11%; 1/9), *I. pavlovsky* (8%; 2/26), *A. testudinarium* (4%; 1/26), and *H. flava* (2%; 1/65) (Figure 1). Only female ticks were positive for the infection in *I. turdus*, *D. taiwanensis*, and *H. flava*, while only one male was positive in *A. testudinarium*. The difference in *Spiroplasma* infection rates between male and female ticks was not statistically significant (Fisher’s exact test). *Spiroplasma*-positive ticks were detected from most of the geographic blocks except for Kanto and Okinawa (Figure 2).

### 3.2. 16S rDNA Genotyping of Spiroplasma in Ticks

A total of 101 amplicons of 16S rDNA were successfully sequenced, resulting in 17 different 16S rDNA alleles (G1–G17) (Table 2). Eight samples failed in sequencing due to mixed signals. Of the 17 alleles, 13 alleles (G3–G8, and G11–G17) were detected in a single tick species. Two alleles (G1 and G10) were detected in two different tick species: G1 from *I. ovatus* and *I. persulcatus* and G10 from *A. testudinarium* and *I. persulcatus*. One allele (G2) was detected in three different tick species: *I. ovatus*, *I. persulcatus*, and *H. kitaokai.* Another allele (G9) was observed in four different tick species: *I. turdus*, *I. persulcatus*, *D. taiwanensis*, and *H. kitaokai*. The detected alleles were classified into the Ixodetis or CCM group in a phylogenetic tree based on the sequences of 16S rDNA (Figure 3). G10 and G17 were clustered with *Spiroplasma* spp. in the CCM group, whereas other alleles were grouped with members in the Ixodetis group. G10 and G17 showed 99.7% and 99.4% sequence identity, respectively, to *S. mirum* (CP006720). Alleles in the Ixodetis group formed a cluster with *S. ixodetis* found in *Ixodes*, *Rhipicephalus,* and *Dermacentor* ticks in other countries and a variety of arthropods such as ladybird, beetle, louse, butterfly, planthopper, and mealybug (Figure 3).

### 3.3. Characterization of Spiroplasma Based on the Sequences of ITS Region, dnaA, and rpoB Genes

To further characterize *Spiroplasma* in ticks, 50 *Spiroplasma*-positive samples were selected based on 16S rDNA genotyping results. The ITS region was amplified in all 16S rDNA alleles, resulting in five different alleles (T1–T5) (Table 3). T1 was the most abundant allele detected in the samples of 10 different 16S rDNA alleles (G1, G2, G4, G8–G10, and G12–G15). Phylogenetic analysis revealed that T4 was clustered with *Spiroplasma* spp. including *S. mirum* in the CCM group, whereas T1-T3 and T5 formed a cluster with *S. ixodetis* reported from butterflies (Figure 4). There was a discrepancy between the 16S rDNA and ITS genotyping results; haplotype SP22 had a 16S rDNA allele (G10) belonging to the CCM group and an ITS allele (T1) belonging to the Ixodetis group. PCR amplification of the *dnaA* and *rpoB* genes were only successful for six and seven 16S rRNA alleles, respectively. ML trees based on *dnaA* and *rpoB* are shown in Appendix A, respectively.

### 3.4. Effect of the Genetic Background on Spiroplasma Infection

Based on the estimation of multicollinearity using VIF, the number of degrees of freedom (Df) was more than 1 for all variables except the year; thus, we calculated the generalized variance inflation factors (GVIFs). The Df is equal to the number of associated coefficients for a GVIF. Therefore, we used GVIF^1^^∕^^2Df^ to make GVIF values comparable among those with different numbers of Df. High collinearity is usually indicated by VIF  >  20. However, multicollinearity analysis using VIF indicated low multicollinearity with all variables (VIF < 5), suggesting that linear regression models would not be influenced by a combination of these variables. Multicollinearity analysis showed that there was a moderate correlation between the predictor variables (season and month; district and city/town) (Appendix A). Both month and city/town variables were excluded from further analysis. Then, MCA was performed to identify associations between the predictor variables. The strongest association was detected between district, species, and season (Appendix A). LMM analysis using the predictor variables (season, year, sex, and species) revealed that the introduction of district as the random effect variable improved the models significantly (*p* ≦ 0.001) (Table 4). Moreover, when tick species was used as the principal predictor, the model for testing *Spiroplasma* infection in ticks was improved (*p* ≦ 1.73 × 10^−75^; Table 5).

The association between *Spiroplasma* 16S rDNA alleles and host tick species was estimated separately using the best-fit LMM. This analysis was applicable to six alleles (G1–G3, G6, G9, and G11). However, the analysis was not appropriate for the other 11 alleles due to the small sample size (less than five). The analysis revealed that the probability of infection with G1 and G11 was significantly associated with *I. ovatus*; however, compared with other tick species, *H. kitaokai* had a significantly higher probability of infection with G9 (Table 6 and Appendix A).

## 4. Discussion

Prior to this study, there was only limited information available on the prevalence and genetic diversity of tick-associated *Spiroplasma* in Japan. In addition to three tick species (*H. kitaokai, I. ovatus*, and *I. persulcatus*) that were previously revealed to harbour *Spiroplasma* [24,32], five additional species, i.e., *A. testudinarium*, *D. taiwanensis*, *H. flava*, *I. pavlovsky*, and *I. turdus*, were found to be infected with *Spiroplasma*, thus expanding our knowledge of the host range of tick-associated *Spiroplasma* in Japan.

The infection rate of *Spiroplasma* ranged from 0% to 84% depending on the tick species. To investigate whether this difference in infection rate is determined by the tick species or other factors, LMM analysis was performed. The results indicated that *Spiroplasma* infection was mainly influenced by the species of ticks but less likely to be influenced by temporal and seasonal factors (Table 5). Although the prevalence of *Spiroplasma* in tick populations has not been well understood, several previous studies reported that the *Spiroplasma* infection rates are variable between populations such as in *I. arboricola*, *I. ricinus,* and *R. decoloratus* [28,43]. A study investigating *Spiroplasma* infection rates in natural *Drosophila* populations in the southwestern United States and northwestern Mexico observed varying infection rates depending on the fly species [44]. In the same study, there was a difference in *Spiroplasma* infection rates in two fly species between the two collection sites. Similarly, in our LMM analysis, the introduction of district as the random effect variable improved the models significantly (Table 4), indicating that the *Spiroplasma* infection status in ticks may be partially influenced by the sampling location.

The highest infection rate was observed in *I. ovatus*; 82% (32/39) of males and 85% (35/41) of females were positive based on PCR amplification of *Spiroplasma* 16S rDNA (Figure 1). Sequencing analysis of PCR amplicons identified 11 *Spiroplasma* alleles in this tick species (Table 3). Furthermore, *H. kitaokai,* the second most infected species (28% (11/40) of males and 42% (16/38) of females), had four different *Spiroplasma* alleles. The association between specific 16S rDNA alleles (G1, G9, and G11) and their host tick species was statistically confirmed (Table 6). The presence of these alleles resulted in the high overall infection rates in *I. ovatus* and *H. kitaokai*. These *Spiroplasma* alleles may have adapted to the tick environment, which is important for symbionts [45]. The transmission of symbionts occurs mainly through the vertical or horizontal route. Vertical transmission involves the dispersal of symbionts and occurs primarily from the mother to offspring. Horizontal transmission occurs via host-to-host contact and acquisition from the environment [45]. The high infection rates observed in *I. ovatus* and *H. kitaokai* suggest the vertical transmission of *Spiroplasma* in these tick species. Symbionts can positively affect the nutrition, reproduction, and defence of their hosts. These positive effects may promote the coexistence or coevolution of symbionts and their hosts [45]. Therefore, it is of particular interest to investigate whether *Spiroplasma* affects tick fitness, as it may help understand the close association between *Spiroplasma* and ticks.

Among the three *Spiroplasma* clades, tick-associated *Spiroplasma* has only been identified in the Ixodetis and CCM groups. In the present study, most of the samples were classified as belonging to the Ixodetis group (*n* = 98), and only three samples were classified as belonging to the CCM group (Figure 3). Considering that most of the *Spiroplasma* species from ticks identified in previous studies belong to the Ixodetis group [21,22,24,25,29,30,43,46], this group of *Spiroplasma* may be widely distributed in the world. On the other hand, there is a lack of information on the geographic distribution and host range of tick-associated *Spiroplasma* in the CCM group. The alleles G10 and G17 obtained in the present study showed high sequence identities (99.7% and 99.4%, respectively) to *S. mirum*, which has been found to cause persistent infection in the mouse brain [47] and neurological deterioration and spongiform encephalopathy in suckling rats [48,49]. Furthermore, several ruminants such as deer, sheep, and goats developed spongiform encephalopathy in a dose-dependent manner when experimentally inoculated with *S. mirum* in their brains [50]. The alleles G10 and G17 were obtained from *A. testudinarium*, *I. pavlovsky*, and *I. persulcatus*, whose primary hosts include domestic and wild ruminants such as cattle and sika deer in Japan [51,52]. Furthermore, *A. testudinarium* and *I. persulcatus* are human-biting species that serve as main vectors for human tick-borne diseases [53,54]. Hence, it is important to investigate the potential of these *Spiroplasma* alleles as agents of human and animal diseases.

The 16S rDNA-based genotyping of 101 *Spiroplasma*-positive samples identified 17 alleles, some of which were observed in more than two different tick species (Table 2). However, further characterization by sequencing additional genes (ITS, *dnaA*, and *rpoB*) divided them into 31 haplotypes, and only one of them (SP24) was observed in two tick species (*A. testudinarium* and *I. persulcatus*) (Table 3). A previous study suggested the possible horizontal transmission of *Spiroplasma* between different ticks and other arthropods, considering that tick-derived *S. ixodetis* did not form a tick species-specific clade [30]. Our results indicated that horizontal transmission among tick species is not common, at least among the tested tick species. However, the fact that certain alleles (G2, G9, and G15) in the Ixodetis group were more related to *Spiroplasma* found in other arthropods than other alleles found in ticks highlights the important role of horizontal transmission between arthropods in the spread of *Spiroplasma* in ticks, as suggested previously [30].

The genes *dnaA* and *rpoB* are frequently used in the detection and characterization of *Spiroplasma* alleles in various arthropods [1,29,36,40,46,55]. In this study, *dnaA* and *rpoB* were not amplified in nearly half of the haplotypes tested (Table 3). This may be attributed to nucleotide mismatches in the primer annealing sites. To understand the genetic diversity of *Spiroplasma* and clarify the mode of horizontal transmission in ticks, further assays using different gene targets and primer sets are necessary. A previous study developed a multi-locus sequence typing method based on five genes (16S rDNA, *rpoB*, *dnaK*, *gyrA*, and *EpsG*) by referring the daft genome of *S. ixodetis* Y32 type [30]. Considering high PCR success rates reported for ticks and other arthropods, the method might be useful to genotype *Spiroplasma* in ticks.

Some species of *Spiroplasma* are known to affect host reproductive systems through mechanisms such as male killing [7,8,9,10]. For instance, *Spiroplasma* kills *Drosophila* males by inducing male X chromosome-specific DNA damage and activating p53-dependent abnormal apoptosis in male embryos [56]. In this study, 49 male ticks and 60 female ticks were infected with *Spiroplasma*, and there was no statistically significant difference for any of the tested tick species (Figure 1). This result is consistent with that of LLM analysis, where sex was not selected as a variable to improve the model of *Spiroplasma* infection in ticks (Table 4). Similarly, two previous studies targeting wild populations of *R. decoloratus* and wild and laboratory populations of *I. arboricola* did not find any association between sex and *Spiroplasma* infection [27,30].

In a previous study, *Spiroplasma* was highly abundant in the salivary glands of *I. ovatus* [23]. It is known that *S. citri*, a plant pathogenic *Spiroplasma*, propagates in the salivary glands of arthropod hosts such as leafhoppers and is released along with the saliva into a new plant during feeding, which leads to transmission from an infected plant to new arthropod hosts [57,58]. Similarly, the presence of *Spiroplasma* in the tick salivary glands may cause horizontal transmission via feeding to unidentified hosts. One recent study reported that the salivary protein components of *Wolbachia/Spiroplasma-*infected spider mites differed from those of uninfected mites [59]. Tick saliva is an important biological material for various processes such as combating host defences, accelerating blood-feeding processes, and facilitating the transmission of pathogens to hosts [60]. Therefore, the effects of *Spiroplasma* on tick physiology and pathogen transmission involving the tick salivary glands should be clarified in future experimental studies.

## 5. Conclusions

*Spiroplasma* is one of the most common symbionts in arthropods; however, only limited data are available on species that infect ticks. This study expanded our knowledge of the host range of tick-associated *Spiroplasma* in Japan. Modelling analysis using tick samples with different infection rates indicated that the host tick species was the primary factor associated with *Spiroplasma* infection. Moreover, the presence of certain alleles that are highly adapted to specific tick species may explain the high infection rates in *I. ovatus* and *H. kitaokai*. A comparison of the alleles suggests that the horizontal transmission of *Spiroplasma* between tick species may not be a frequent event. Further studies are required to understand the transmission cycle of *Spiroplasma* species in wild tick populations and their roles in ticks.

## Figures and Tables

**Figure 1 microorganisms-09-00333-f001:**
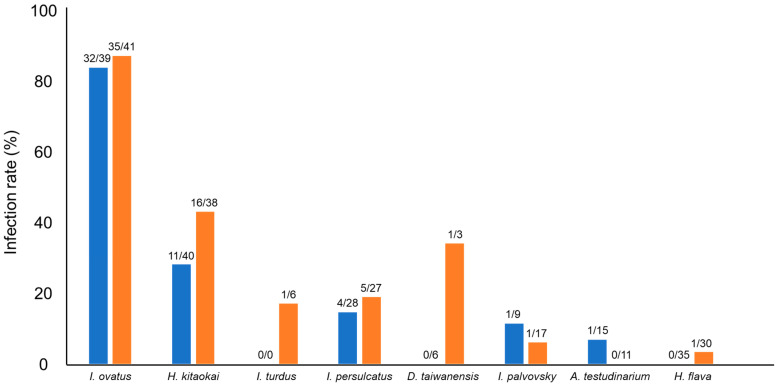
*Spiroplasma*-positive rates of different tick species. Blue and orange bars represent male and female ticks, respectively. The numbers at the top of the bars indicate the number of *Spiroplasma*-positive ticks/number of tested ticks.

**Figure 2 microorganisms-09-00333-f002:**
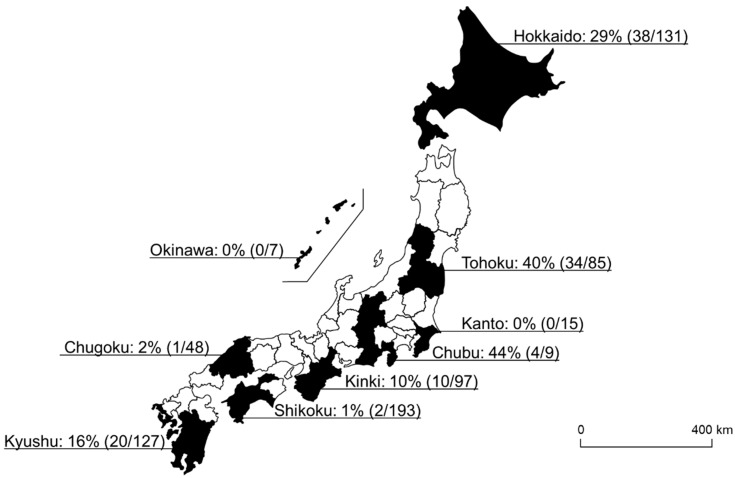
A map of Japan showing the *Spiroplasma*-positive rate of each geographical block. The numbers in the parentheses refer to the number of *Spiroplasma*-positive ticks/number of tested ticks.

**Figure 3 microorganisms-09-00333-f003:**
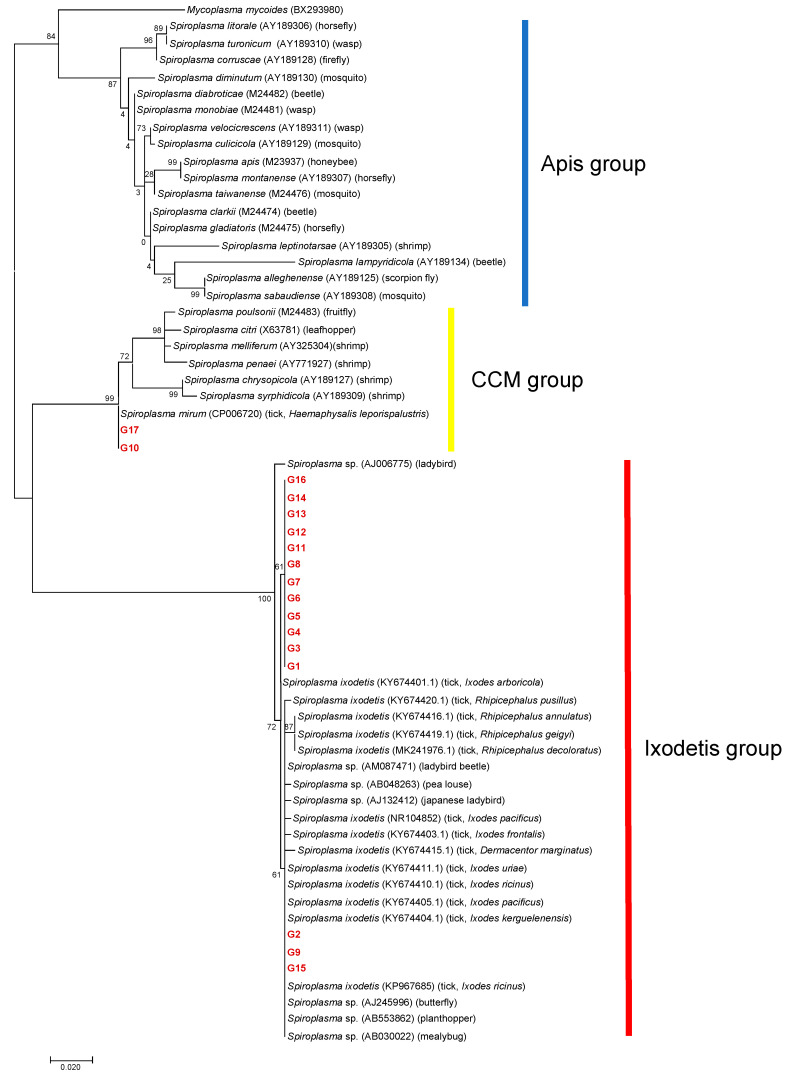
A phylogenetic tree based on the sequences of 16S rDNA. The analysis was performed using a maximum-likelihood method based on the Hasegawa–Kishino–Yano model with bootstrap tests of 1000 replicates in MEGA7. A discrete Gamma distribution was used to model evolutionary rate differences among sites (five categories (+G, parameter = 0.2496)). The sequences obtained in this study are included with allele names provided in Table 2 and are shown in red. The sequences of other *Spiroplasma* species were retrieved from GenBank. The host is indicated in the parenthesis for each *Spiroplasma* sequence.

**Figure 4 microorganisms-09-00333-f004:**
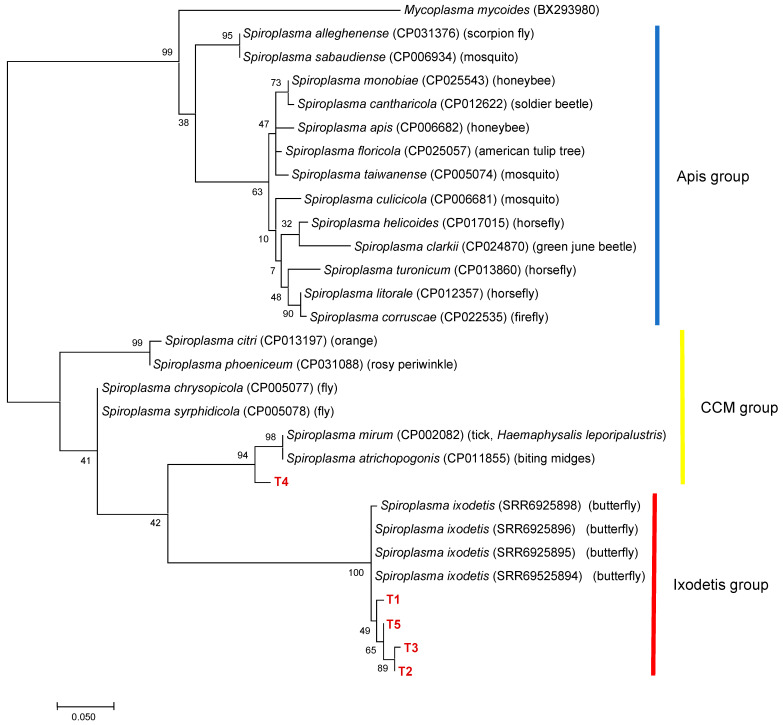
A phylogenetic tree based on the sequences of ITS region. The analysis was performed using a maximum-likelihood method based on the Tamura 3-parameter model. A discrete Gamma distribution was used to model evolutionary rate differences among sites (five categories (+G, parameter = 0.2599)) with bootstrap tests of 1000 replicates in MEGA7. The sequences obtained in this study are included with allele names provided in Table 3 and are shown in red. The sequences of other *Spiroplasma* species were retrieved from GenBank.

**Table 1 microorganisms-09-00333-t001:** Primers used in the present study.

Primer	Sequence (5’-3’)	Target Gene	Annealing Temperature (°C)	Purpose	Amplicon Size (bp)	Reference
spi_f1	GGGTGAGTAACACGTATCT	16S rDNA	60	PCR	1028	[13]
spi_r3	CCTTCCTCTAGCTTACACTA					
16S_s1	ACCTTACCAGAAAGCCACGG	16S rDNA	NA	Sequencing	NA	This study
16S_s2	AGACCTTCATCAGTCACGCG	16S rDNA	NA	Sequencing	NA	This study
16S_s3	GTAATATGTGCCAGCAGCCG	16S rDNA	NA	Sequencing	NA	This study
16S_s4	ACCGCATTCTCCATCAGCTT	16S rDNA	NA	Sequencing	NA	This study
SP-ITS-JO4	GCCAGAAGTCAGTGTCCTAACCG	ITS1	56	PCR	301	[13]
SP-ITS-N55	ATTCCAAGCCATCCACCATACG					
SRdnaAF1	GGAGAYTCTGGAYTAGGAAA	*dnaA*	52	PCR	515	[36]
SRdnaAR1	CCYTCTAWYTTTCTRACATCA					
RpoBF1	ATGGATCAAACAAATCCATTAGCAGA	*rpoB*	60	PCR	1703	[36]
RpoBR2	GCATGTAATTTATCATCAACCATGTGTG					
RpoB_s1	TGACCATTACTACGAGCAATAACA	*rpoB*	NA	Sequencing	NA	This study
RpoB_s2	CCCCTGTTTTTGATGGTGCA	*rpoB*	NA	Sequencing	NA	This study

NA, not applicable.

**Table 2 microorganisms-09-00333-t002:** *Spiroplasma* 16S rDNA alleles and their geographic origins and host tick species.

16S rDNA Allele	Tick Species	No. of Positive/No. of Tested (%)								
		Hokkaido	Tohoku	Kanto	Chubu	Kinki	Chugoku	Sikoku	Kyushu	Okinawa
G1	*I. ovatus*	21/44 (48)	2/32 (6)	-	0/4 (0)	-	-	-	-	-
G1	*I. persulcatus*	1/39 (3)	0/8 (0)	-	0/4 (0)	0/4 (0)	-	-	-	-
G2	*H. kitaokai*	-	0/5 (0)	-	-	2/12 (17)	-	0/36 (0)	0/45 (0)	-
G2	*I. ovatus*	1/44 (2)	0/32 (0)	-	0/4 (0)	-	-	-	-	-
G2	*I. persulcatus*	3/39 (7)	0/8 (0)	-	0/4 (0)	0/4 (0)	-	-	-	-
G3	*I. ovatus*	3/44 (7)	1/32 (3)	-	3/4 (75)	-	-	-	-	-
G4	*I. ovatus*	1/44 (2)	0/32 (0)	-	0/4 (0)	-	-	-	-	-
G5	*I. ovatus*	1/44 (2)	0/32 (0)	-	0/4 (0)	-	-	-	-	-
G6	*I. ovatus*	3/44 (7)	1/32 (3)	-	1/4 (25)	-	-	-	-	-
G7	*I. ovatus*	1/44 (2)	1/32 (3)	-	0/4 (0)	-	-	-	-	-
G8	*I. ovatus*	1/44 (2)	0/32 (0)	-	0/4 (0)	-	-	-	-	-
G9	*D. taiwanensis*	-	0/1 (0)	-	˗	1/4 (25)	-	-	0/4 (0)	-
G9	*H. kitaokai*	˗	0/5 (0)	-	˗	3/12 (25)	-	2/36 (6)	18/45 (40)	-
G9	*I. persulcatus*	0/39 (0)	0/8 (0)	-	0/4 (0)	4/4 (100)	-	-	-	-
G9	*I. turdus*	-	-	-	-	0/2 (0)	-	0/2 (0)	1/2 (50)	-
G10	*A. testudinarium*	-	˗	-	˗	-	1/9 (11)	0/15 (0)	0/2 (0)	-
G10	*I. persulcatus*	0/39 (0)	1/8 (13)	-	0/4 (0)	0/4 (0)	-	-	-	-
G11	*I. ovatus*	0/44 (0)	16/32 (50)	-	0/4 (0)	-	-	-	-	-
G12	*I. ovatus*	0/44 (0)	2/32 (6)	-	0/4 (0)	-	-	-	-	-
G13	*I. pavlovsky*	1/26 (4)	-	-	-	-	-	-	-	-
G14	*H. kitaokai*	-	0/5 (0)	-	-	0/12 (0)	-	0/36 (0)	1/45 (2)	-
G15	*H. kitaokai*	-	1/5 (20)	-	-	0/12 (0)	-	0/36 (0)	0/45 (0)	-
G16	*I. ovatus*	0/44 (0)	1/32 (3)	-	0/4 (0)	-	-	-	-	-
G17	*I. pavlovsky*	1/26 (4)	-	-	-	-	-	-	-	-

**Table 3 microorganisms-09-00333-t003:** Multi-locus sequence typing of *Spiroplasma* in ticks.

*Spiroplasma* Haplotype	16S rDNA	ITS	*dnaA*	*rpoB*	Tick Species
SP1	G1	T3	A1	B1	*I. ovatus*
SP2	G1	T1	-	-	*I. persulcatus*
SP3	G2	T1	A1	B4	*H. kitaokai*
SP4	G2	T1	A1	-	*H. kitaokai*
SP5	G2	T2	-	-	*I. ovatus*
SP6	G2	T1	A2	B1	*I. persulcatus*
SP7	G2	T1	A2	B7	*I. persulcatus*
SP8	G2	T1	-	-	*I. persulcatus*
SP9	G3	T2	-	-	*I. ovatus*
SP10	G4	T1	A2	B3	*I. ovatus*
SP11	G5	T3	A2	B3	*I. ovatus*
SP12	G6	T2	-	-	*I. ovatus*
SP13	G7	T2	A1	-	*I. ovatus*
SP14	G8	T1	A2	B3	*I. ovatus*
SP15	G9	T1	A2	B2	*D. taiwanensis*
SP16	G9	T1	A2	B4	*H. kitaokai*
SP17	G9	T1	A2	B7	*H. kitaokai*
SP18	G9	-	A2	B7	*H. kitaokai*
SP19	G9	T1	-	-	*I. persulcatus*
SP20	G9	T1	A1	-	*I. persulcatus*
SP21	G9	T1	A1	B7	*I. persulcatus*
SP22	G9	T5	-	B6	*I. persulcatus*
SP23	G9	T1	-	B5	*I. turdus*
SP24	G10	T1	-	-	*A. testudinarium*
	G10	T1	-	-	*I. persulcatus*
SP25	G11	T2	-	-	*I. ovatus*
SP26	G12	T1	-	-	*I. ovatus*
SP27	G13	T1	-	-	*I. pavlovsky*
SP28	G14	T1	-	-	*H. kitaokai*
SP29	G15	T1	-	-	*H. kitaokai*
SP30	G16	T2	-	-	*I. ovatus*
SP31	G17	T4	-	-	*I. pavlovsky*

-, Not amplified.

**Table 4 microorganisms-09-00333-t004:** LMM to test the correlation between each predictor with *Spiroplasma* infection using district as the random effect variable.

Model	Predictor Variable	Random Variable	AIC	BIC	logLik	Dev	Chisq	Df	Pr (>Chisq)	
**M1-1**	Species	No	99.33	195.26	−28.67	57.33	NA	NA	NA	
**M1-2**	Species	District	74.43	174.93	−15.22	30.43	26.90	1	2.14 × 10^−7^	***
**M2-1**	Year	No	467.30	481.00	−230.65	461.30	NA	NA	NA	
**M2-2**	Year	District	459.39	477.67	−225.70	451.39	9.90	1	0.00164998	***
**M3-1**	Sex	No	495.23	513.50	−243.61	487.23	NA	NA	NA	
**M3-2**	Sex	District	451.24	474.08	−220.62	441.24	45.99	1	1.19 × 10^−11^	***
**M4-1**	Season	No	538.56	556.83	−265.28	530.56	NA	NA	NA	
**M4-2**	Season	District	465.98	488.82	−227.99	455.98	74.58	1	5.83 × 10^−18^	***

NA: Not applicable; AIC: Akaike information criterion; BIC: Bayesian information criterion; logLik: log-likelihood; ChiSq: ANOVA Chi-square value; Dev: Deviance of the model; Df: Chi-square degrees of freedom; Pr(>Chisq): ANOVA *p* value. The level of significance was marked as *** if *p* < 0.0001 and not marked if *p* > 0.05.

**Table 5 microorganisms-09-00333-t005:** Effect of several variables on the probability of *Spiroplasma* infection in the LMM.

Model	Predictor Variable	Random Variable	AIC	BIC	LogLik	Deviance	Chisq	Df	Pr (>Chisq)	
M5	NO	District	464.22	477.92	−229.11	458.22	NA	NA	NA	
M7	Year	District	459.39	477.67	−225.70	451.39	6.82	1	0.00899482	**
M8	Season	District	451.24	474.08	−220.62	441.24	10.16	1	0.00143586	**
M9	Sex	District	465.98	488.82	−227.99	455.98	0.00	0	NA	
M6	Species	District	74.43	174.93	−15.22	30.43	425.55	17	8.34 × 10^−80^	***
M10	Season + Species	District	71.83	181.47	−11.92	23.83	6.60	2	0.03694614	*
M11	Species + Season	District	71.83	181.47	−11.92	23.83	0.00	0	NA	
M12	Species + Season + Sex	District	69.87	188.64	−8.93	17.87	5.97	2	0.05065574	.

NA: Not applicable; AIC: Akaike information criterion; BIC: Bayesian information criterion; logLik: log-likelihood; ChiSq: ANOVA Chi-square value; Dev: Deviance of the model; Df: Chi-square degrees of freedom; Pr(>Chisq): ANOVA *p* value. The level of significance was marked as *** if *p* < 0.0001 and not marked if *p* > 0.05.

**Table 6 microorganisms-09-00333-t006:** Association between *Spiroplasma* 16S rDNA alleles and tick species.

16S rDNA Allele	Tick Species (No. of Positive Samples)	Significance
G1	*I. ovatus* (*n* = 23), *I. persulcatus* (*n* = 1)	*I. ovatus*
G2	*H. kitaokai* (*n* = 2), *I. ovatus* (*n* = 1), *I. persulcatus* (*n* = 3)	Not significant
G3	*I. ovatus* (*n* = 7)	Not significant
G4	*I. ovatus* (*n* = 1)	NA
G5	*I. ovatus* (*n* = 1)	NA
G6	*I. ovatus* (*n* = 5)	Not significant
G7	*I. ovatus* (*n* = 2)	NA
G8	*I. ovatus* (*n* = 1)	NA
G9	*D. taiwanensis* (*n* = 1), *H. kitaokai* (*n* = 23), *I. turdus* (*n* = 1), *I. persulcatus* (*n* = 4)	*H. kitaokai*
G10	*A. testudinarium* (*n* = 1), *I. persulcatus* (*n* = 1)	NA
G11	*I. ovatus* (*n* = 16)	*I. ovatus*
G12	*I. ovatus* (*n* = 2)	NA
G13	*I. pavlovsky* (*n* = 1)	NA
G14	*H. kitaokai* (*n* = 1)	NA
G15	*H. kitaokai* (*n* = 1)	NA
G16	*I. ovatus* (*n* = 1)	NA
G17	*I. pavlovsky* (*n* = 1)	NA

NA: not applicable.

## Data Availability

The data presented in this study are available in the DNA Data Bank of Japan (DDBJ) (http://www.ddbj.nig.ac.jp) and the accession numbers are available in the text.

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
