# Peer review of "Spiroplasma Infection among Ixodid Ticks Exhibits Species Dependence and Suggests a Vertical Pattern of Transmission"

_microorganisms, 2021, doi:10.3390/microorganisms9020333_

Round 1

Reviewer 1 Report

The authors present an interesting study on the diversity of Spiroplasma in Japanese ticks. Spiroplasma is one of the most common endosymbionts found in arthropods, and some Spiroplasma species, such S. ixodetis, are reproductive parasites, inducing death of male during early embryo development, or defensive symbionts, protecting hosts against environmental stresses. Such manipulations have then profound effects on host phenotypes and this type of endosymbionts are now regarded as very important cryptic drivers of arthropod ecology and evolution. In this study, the authors identified 8 tick species infected by Spiroplasma, and further characterized infection prevalence and genetic diversity through multi-locus sequencing. A substantial diversity of Spiroplasma genotypes were identified, and based on the association with tick species, the authors concluded that tick species is a key factor explaining Spiroplasma infection diversity. The rationale of the study is clear and the ms is well written. I have only one major comment, and a couple of minor comments, as I detailed below.

Major comment:

While the authors clearly indicated that Spiroplasma is common in arthropods, they only use a few DNA sequences from public database in their phylogenetic analyses. Indeed, there are plethora of 16S Spiroplasma sequences (many are from ticks sampled worldwide) in GenBank, and I was wondering if none is identical to the G8, G12, G6, G3, etc, genotypes showed at Figure 3. In addition, the study by Binetruy et al (2019, TTBD, reference 38 in the ms) produced rpoB sequences from Spiroplasma of a wide diversity of arthropods, including insects and a dozen of tick species. Why not use these sequences for comparisons, especially those from other tick species? It will be helpful to test the specificity of Spiroplasma genotypes to their respective tick species. Also note that it is not true that there is no ITS sequence of S. ixodetis in GenBank (as stated at line 236) since there are several complete genomes currently available (eg some are from the African monarch butterfly Danaus chrysippus, cf. Martin et al, PLoS Biol, 2020). These S. ixodetis sequences could be included in the ITS dataset. Overall, the use of additional sequences is necessary if the authors want to compare Spiroplasma infections between tick species. Without this, I feel that this may lead to a partial understanding of Spiroplasma infection dynamics. Indeed, the authors stated that horizontal transfer between tick species is not frequent. I agree with this because only few Spiroplasma genotypes were shared by different tick species. However, it seems also obvious at Figure 3 that some Spiroplasma genotypes of certain tick species are more related to Spiroplasma genotypes of insects that to those of other tick species. The study by Binetruy et al (2019, TTBD, reference 38 in the ms) well explain with this pattern through repeated horizontal transfers between arthropod species, including ticks and insects. So, I think that it is important to make clear that interspecific horizontal transfer of Spiroplasma may be rare but are actually important in establishing infections in other tick species.

Minor comments:

  • Line 53: Reference #38 (Binetruy et al 2019, TTBD) that described a dozen of Spiroplasma genotypes in a diversity of tick species must be cited here.
  • Lines 127-140: Again, why not use the multi-locus strain typing method (Binetruy et al 2019, TTBD) specifically developed for Spiroplasma found in ticks?
  • Lines 314-316: This sentence is not exactly true since the prevalence of Spiroplasma have been also extensively examined in several geographical population of I. ricinus, I. arboricola and R. decoloratus (eg Duron et al 2017, Molecular Ecology) – it showed that infection prevalence was highly variable between tick populations.

Other comments:

  • Line 51: fruit fly natural populations, rather than wild fly populations
  • Line 82 and further: Ticks were collected by flagging but there is no details on stages. Were all specimens adult? No nymph or larva was examined or collected?

Author Response

The authors present an interesting study on the diversity of Spiroplasma in Japanese ticks. Spiroplasma is one of the most common endosymbionts found in arthropods, and some Spiroplasma species, such S. ixodetis, are reproductive parasites, inducing death of male during early embryo development, or defensive symbionts, protecting hosts against environmental stresses. Such manipulations have then profound effects on host phenotypes and this type of endosymbionts are now regarded as very important cryptic drivers of arthropod ecology and evolution. In this study, the authors identified 8 tick species infected by Spiroplasma, and further characterized infection prevalence and genetic diversity through multi-locus sequencing. A substantial diversity of Spiroplasma genotypes were identified, and based on the association with tick species, the authors concluded that tick species is a key factor explaining Spiroplasma infection diversity. The rationale of the study is clear and the ms is well written. I have only one major comment, and a couple of minor comments, as I detailed below.

Answer: We would like to appreciate the reviewer’s corrections and suggestions on the manuscript. All the comments were reflected in the revision.

 I was wondering if none is identical to the G8, G12, G6, G3, etc, genotypes showed at Figure 3. In addition, the study by Binetruy et al (2019, TTBD, reference 38 in the ms) produced rpoB sequences from Spiroplasma of a wide diversity of arthropods, including insects and a dozen of tick species. Why not use these sequences for comparisons, especially those from other tick species? It will be helpful to test the specificity of Spiroplasma genotypes to their respective tick species. Also note that it is not true that there is no ITS sequence of S. ixodetis in GenBank (as stated at line 236) since there are several complete genomes currently available (eg some are from the African monarch butterfly Danaus chrysippus, cf. Martin et al, PLoS Biol, 2020).

Answer: We agree with the comment. We added sequence data of other tick species reported in the previous study (Binetruy et al (2019, TTBD) to phylogenetic trees based on 16S rDNA (Fig. 3) and rpoB (Fig S1).

It is unfortunate that the authors of the paper (Martin et al, PLoS Biol, 2020) deposited illumina raw reads and mitochondrial genomes of the butterflies but they did not deposit genome data of S. ixodetis they assembled in the public database. Hence, we downloaded the raw illumine reads from SRA database and generated ITS sequences of four S. ixodestis from butterflies. We explained this in Materials and Methods and revised Fig. 4.

Line 151-158:

The reference sequences of ITS region of S. ixodetis were obtained by de novo assembly of Illumina raw reads of Spiroplasma-infected African monarch butterfly Danaus chrysippus deposited in the sequence read archives (SRA) of the NCBI with accession nos. of SRX3872086 and SRX3872088-SRX3872090 [37] using CLC Genomics Workbench v 20.0.4 (Qiagen, Hilden, Germany).

Line 241-245:

Phylogenetic analysis revealed that T4 was clustered with Spiroplasma spp. including S. mirum in the CCM group, whereas T1-T3 and T5 formed a cluster with S. ixodetis reported from butterflies (Figure 4). There was a discrepancy between the 16S rDNA and ITS genotyping results; haplotype SP22 had a 16S rDNA allele (G10) belonging to the CCM group and an ITS allele(T1) belonging to the Ixodetis group.

Indeed, the authors stated that horizontal transfer between tick species is not frequent. I agree with this because only few Spiroplasmagenotypes were shared by different tick species. However, it seems also obvious at Figure 3 that some Spiroplasma genotypes of certain tick species are more related to Spiroplasma genotypes of insects that to those of other tick species. The study by Binetruy et al (2019, TTBD, reference 38 in the ms) well explain with this pattern through repeated horizontal transfers between arthropod species, including ticks and insects. So, I think that it is important to make clear that interspecific horizontal transfer of Spiroplasma may be rare but are actually important in establishing infections in other tick species.

Answer: We agree with the reviewer’s comment. We think that there was a lack of explanation in Results and Discussion on horizontal transmission. We explained the co-clustering between S. ixodetis from ticks and other arthropods in Results. We also highlighted the important role of horizontal transmission in the spread of Spiroplasma in ticks in Discussion.

Line 203-206:

Alleles in the Ixodetis group formed a cluster with S. ixodetis found in Ixodes, Rhipicephalus and Dermacentor ticks in other countries and a variety of arthropods such as ladybird, beetle, louse, butterfly, planthopper, and mealybug (Figure 3). 

Line 371-374:

However, the fact that certain alleles (G2, G9 and G15) in the Ixodetis group were more related to Spiroplasma found in other arthropods than other alleles found in ticks highlights the important role of horizontal transmission between arthropods in the spread of Spiroplasma in ticks as suggested previously [30].

  • Line 53: Reference #38 (Binetruy et al 2019, TTBD) that described a dozen of Spiroplasma genotypes in a diversity of tick species must be cited here.

Answer: We agree with the comment and added this reference in this sentence.

Lines 127-140: Again, why not use the multi-locus strain typing method (Binetruy et al 2019, TTBD) specifically developed for Spiroplasma found in ticks?

Answer: We appreciate the comment. When we initiated the present study, the paper reporting the multi-locus typing method was not published. After we read this very informative paper, we thought that their PCR might amplify more than the PCR we used.

One advantage of the PCR we employed is that it can amplify longer region (1703 bp) than the PCR (628 bp). Meanwhile, when we compared the results of 16S rDNA and rpoB genotyping, we found that 16S rDNA had higher resolution than ropB. For examples, some ropB alleles (B1, B3, and B7) were further divided into two or three different 16S rDNA alleles. So we concluded that application of new PCR might increase PCR positives but it might not be related to genetic separation of Spiroplasma found in this study. Nonetheless, considering the high PCR success rates of the new PCR, the method should be applicable for wide range of samples. We included the following discussion in the revision:

Line 378-384:

To understand the genetic diversity of Spiroplasma and clarify the mode of horizontal transmission in ticks, further assays using different gene targets and primer sets are necessary. A previous study developed a multi-locus sequence typing method based on five genes (16S rDNA, rpoB, dnaK, gyrA, and EpsG) by referring the daft genome of S. ixodetis Y32 type [30]. Considering high PCR success rates reported for both ticks and other arthropods, the method might be useful to genotype Spiroplasma in ticks.

Lines 314-316: This sentence is not exactly true since the prevalence of Spiroplasma have been also extensively examined in several geographical population of I. ricinus, I. arboricola and R. decoloratus (eg Duron et al 2017, Molecular Ecology) – it showed that infection prevalence was highly variable between tick populations.

Answer: We agree with the comment. We revised the text as follows:

Line 316-319:

Although the prevalence of Spiroplasma in tick populations has not been well understood, several previous studies reported that the Spiroplasma infection rates are variable between populations such as in I. arboricola, I. ricinus, and R. decoloratus[28,43].

Line 51: fruit fly natural populations, rather than wild fly populations

Answer: We revised the text according to the suggestion.

Line 82 and further: Ticks were collected by flagging but there is no details on stages. Were all specimens adult? No nymph or larva was examined or collected?

Answer: We examined only adult ticks in this study. This information is provided in the Materials and Methods as follows:

Line 94:

A total of 712 adult ticks from four genera were examined in this study.

Reviewer 2 Report

This study examined ixodid ticks collected from different regions in Japan for infection rates of Spiroplasma by PCR targeting 16S rDNA, ITS, dnaA, and rpoB genes, sequencing, and characterizing the genotypes and phylogenic trees and conducted a linear mixed model with Chi-square testing via ANOVA to examine extrinsic factors including sampling location and seasonality. variations, and the intrinsic factors were tick species and sex. The diversity of the 712 ticks collected included the genera Amblyomma (2 species); Dermacento (1 species); Haemaphysalis (10 species); and Ixodes (7 species). Their results suggested that tick species was the primary factor associated with Spiroplasma infection and that certain Spiroplasma genotypes are highly adapted to specific tick species such as Ixodes ovatus and Haemaphysalis kitaokai.

In general, this is an interesting study and the data appears to be acquired correctly although it is not clear if the spiroplasmas were isolated from whole individual ticks or from salivary glands. Moreover, according to this study a genotype is associated as a different branch on a phylogenetic tree. I would not call this a genotype as it could simply be a difference of several nucleotides with no biological relevance. Although the complexity of the tick species, sex, location, collection date, etc. was noteworthy, in the final analysis, vertical transmission may be the major factor. A key factor missing appears to be the host from which the tick was collected from.

Specific comments:

Ln 5. 104-107. It is unclear how DNA was isolated from ticks based on “homogenised in 100 μL of high

glucose Dulbecco's modified Eagle's medium (Gibco, Life Technologies, Grand Island,NY, USA) using Micro SmashTM MS100R (TOMY, Tokyo, Japan) for 30 s at 3,000 rpm as previously reported [35]. Nakao et al. (2011) cultured rickettsia strains in bovine aorta endothelial cells as described previously [12] and subjected to

DNA extraction using the Nucleospin Tissue Kit (Macherey-Nagel, Duren, Germany). Please explain this discrepancy. The methodology described was taken exactly from [34] (Thu et al. 2019). Were the spiroplasma isolated from salivary glands as was described in [23 Qui et al. 2014)?

Ln 120. Table 1 Were primers for sp_f1/spi_r3 and SP-ITS-JO4/SP-ITS-N55 from [17}] or [13]?

Ln 208 Table 2. It is not clear what is the basis for 16s rDNA genotypes G1-G17. No descriptions are provided other than what is included in Table 1. The genotypes appear to be taken from Figure 3 from the CCM and Ixodetis groups. I am not sure this is the correct use of the term “genotype” based solely on the phylogenetic tree of the 16S rDNA.

Ln 246. Table 2 and Fig. 4. Ditto for ITS groups T1-T5; Spiroplasma haplotypes SP1-31; dnaA A1—A2; and rpoB B1-7.

Author Response

In general, this is an interesting study and the data appears to be acquired correctly although it is not clear if the spiroplasmas were isolated from whole individual ticks or from salivary glands. Moreover, according to this study a genotype is associated as a different branch on a phylogenetic tree. I would not call this a genotype as it could simply be a difference of several nucleotides with no biological relevance. Although the complexity of the tick species, sex, location, collection date, etc. was noteworthy, in the final analysis, vertical transmission may be the major factor. A key factor missing appears to be the host from which the tick was collected from.

Answer: We would like to appreciate the reviewer’s suggestions on our manuscript. In this study, we used DNA extracted from whole individual ticks. We clarified it in the revised text. The word “genotype” was replaced with “allele” throughout the manuscript. We collected ticks from vegetations but not from animals.

Ln 5. 104-107. It is unclear how DNA was isolated from ticks based on “homogenised in 100 μL of high glucose Dulbecco's modified Eagle's medium (Gibco, Life Technologies, Grand Island,NY, USA) using Micro SmashTM MS100R (TOMY, Tokyo, Japan) for 30 s at 3,000 rpm as previously reported [35]. Nakao et al. (2011) cultured rickettsia strains in bovine aorta endothelial cells as described previously [12] and subjected to DNA extraction using the Nucleospin Tissue Kit (Macherey-Nagel, Duren, Germany). Please explain this discrepancy. The methodology described was taken exactly from [34] (Thu et al. 2019). Were the spiroplasma isolated from salivary glands as was described in [23 Qui et al. 2014)?

Answer: We apologize for unclear explanations. Whole tick bodies were firstly homogenized in 100 μL of high glucose Dulbecco's modified Eagle's medium (Gibco, Life Technologies, Grand Island,NY, USA) using Micro SmashTM MS100R (TOMY, Tokyo, Japan) for 30 s at 3,000 rpm. The half volume of the homogenates were subjected to DNA extraction using a blackPREP Tick DNA/RNA Kit (Analytikjena, Germany) according to the manufacturer’s instructions. We clarified this in the revised text as follows:

Line 103-110:

The procedures for DNA extraction from individual ticks have been reported previously [34]. In brief, the surface of tick bodies was individually washed with 70% ethanol and sterilised phosphate-buffered solution (PBS). The whole tick bodies were homogenised in 100 μL of high-glucose Dulbecco's modified Eagle's medium (Gibco, Life Technologies, Grand Island, NY, USA) using Micro SmashTM MS100R (TOMY, Tokyo, Japan) for 30 s at 3,000 rpm. DNA was extracted from 50 μL of the tick homogenate using the blackPREP Tick DNA/RNA Kit (Analytik Jena, Jena, Germany) according to the manufacturer’s protocol.

Ln 120. Table 1 Were primers for sp_f1/spi_r3 and SP-ITS-JO4/SP-ITS-N55 from [17}] or [13]?

Answer: We agree with the comment. [13] is correct.

Ln 208 Table 2. It is not clear what is the basis for 16s rDNA genotypes G1-G17. No descriptions are provided other than what is included in Table 1. The genotypes appear to be taken from Figure 3 from the CCM and Ixodetis groups. I am not sure this is the correct use of the term “genotype” based solely on the phylogenetic tree of the 16S rDNA.

Answer: We agree with the comment. I changed the word “genotype” to “allele”. We also include the following sentence in the figure legend.

Line 233:

The sequences obtained in this study are included with allele names provided in Table 2 and are shown in red. 

Ln 246. Table 2 and Fig. 4. Ditto for ITS groups T1-T5; Spiroplasma haplotypes SP1-31; dnaA A1—A2; and rpoB B1-7.

Answer: We agree with the comment. We included the following sentences in each figure legend:

Line 258,259:

The sequences obtained in this study are included with allele names provided in Table 3 and are shown in red.

Reviewer 3 Report

The manuscript deals with the ticks harboring Spiroplasma. Ticks have been known to harbor Spirochaetes causing several diseases, a big rampant example of a painful disease is the Lyme disease. Therefore these studies are highly relevant to the scientific community where their impact goes beyond the science.

The manuscript is well written, there are a few formatting errors and few minor changes that need to be looked into 

  1. The title needs to be rephrased.
  2. Line # 22, 39- Wall-less need to be rephrased.
  3. Line #24- it has to be explicitly explained that the role of Spiroplasma in Ticks has not been elucidated yet. The sentence is confusing.
  4. Line #55- grammar correction needed
  5. There are some space formatting errors to be looked into
  6. Mentioning the limitations of the study is helpful. 
  7. Were the Spiroplasma collected and cultured prior to 16S? That needs to be specified. You could mention the medium or the method.

Author Response

The manuscript deals with the ticks harboring Spiroplasma. Ticks have been known to harbor Spirochaetes causing several diseases, a big rampant example of a painful disease is the Lyme disease. Therefore these studies are highly relevant to the scientific community where their impact goes beyond the science.

Answer: We would like to appreciate the reviewer’s corrections and suggestions on the manuscript. The comments were reflected in the revision.

The title needs to be rephrased.

Answer: We changed the title as follows:

Spiroplasma infection among ixodid ticks exhibits species dependence and suggests a vertical pattern of transmission.

Line # 22, 39- Wall-less need to be rephrased.

Answer: We replace the words “wall-less gram-positive bacteria” with “gram-positive bacteria without cell walls”.

Line 24,41:

Members of the genus Spiroplasma are gram-positive bacteria without cell walls.

Line #24- it has to be explicitly explained that the role of Spiroplasma in Ticks has not been elucidated yet. The sentence is confusing.

Answer: We rephrased the sentence as follows:

Line 26:

Ticks also harbour Spiroplasma but their role has not been elucidated yet.

Line #55- grammar correction needed

Answer: The sentence was corrected as follows:

Line 57:

Ticks have long been studied since they transmit a variety of pathogens to humans and animals.

There are some space formatting errors to be looked into

Answer: We agree with the comment. All the space formatting errors were corrected.

Mentioning the limitations of the study is helpful.

Answer: We appreciate this suggestion. We included the following sentences in the Discussion to explain the limitations of this study.

Line 378-384:

To understand the genetic diversity of Spiroplasma and clarify the mode of horizontal transmission in ticks, further assays using different gene targets and primer sets are necessary. A previous study developed a multi-locus sequence typing method based on five genes (16S rDNA, rpoB, dnaK, gyrA, and EpsG) by referring the daft genome of S. ixodetis Y32 type [30]. Considering high PCR success rates reported for both ticks and other arthropods, the method might be useful to genotype Spiroplasma in ticks.

Line 402-406:

Tick saliva is an important biological material for various processes such as combating host defences, accelerating blood-feeding processes, and facilitating the transmission of pathogens to hosts [61]. Therefore, the effects of Spiroplasma on tick physiology and pathogen transmission involving the tick salivary glands should be clarified in future in vivo studies.

Were the Spiroplasma collected and cultured prior to 16S? That needs to be specified. You could mention the medium or the method.

Answer: We apologize for unclear explanation. In this study, the whole tick bodies were used for extraction DNA and we didn’t use cultured Spiroplasma.

Round 2

Reviewer 1 Report

The revised manuscript is improved over the original submission: the authors have added some key details (and novel analyses) in their ms and I appreciate this effort. Thanks for that. There is no additional item that requires further clarification.

Author Response

Thank you very much for reviewing our paper. We are relieved to hear that there is no additional item that requires further clarification.